# Disrupting the Status-Quo of Organisational Board Composition to Improve Sustainability Outcomes: Reviewing the Evidence

**Kim Beasy** * **and Fred Gale** 

College of Arts, Law & Education, University of Tasmania, Hobart, TAS 7005, Australia; Fred.Gale@utas.edu.au
* Correspondence: Kim.Beasy@utas.edu.au; Tel.: +61-4-363-243-651

**Abstract:** Sustainability, conceptualised as the integration of economic, social and environmental values, is the 21st century imperative that demands that governments, business and civil society actors improve their existing performance, yet improvement has been highly fragmented and unacceptably slow. One explanation for this is the lack of diversity on the boards of organisations that perpetuates a narrow business, economic and legal mindset rather than the broader integrated values approach that sustainability requires. This paper presents a systematic review of the literature investigating how board diversity affects the sustainability performance of organisations. Our review uncovers evidence of relationships between various attributes of the diversity of board members and sustainability performance, though over-reliance on quantitative methodologies of studies reviewed means explanations for the observed associations are largely absent. Limited measures of sustainability performance and narrow definitions of diversity, focused predominantly on gender, were also found. Important implications from the study include the need for policy responses that ensure boards are diversely composed. We identify that more qualitative investigations into the influence of a broader range of types of board diversity on sustainability performance is needed, along with studies that focus on public sector boards, and research that takes an intersectional understanding of diversity.

**Keywords:** organisational boards; sustainability performance; diversity; gender

## 1. Introduction

In the context of the bushfire emergency in Australia which dramatically illustrates the climate crisis scientists have been warning the world about for 30 years, it is pertinent to recall the comments of a former Australian Prime Minister, Bob Hawke, who in 1989 issued Australia's first major statement on sustainable development, entitled 'Our Country, Our Future'. He stated that 'the crux of the issue in implementing sustainable development is establishing mechanisms that ensure an integration of economic and environmental considerations both now and in the future' [1]. As is now evident, since then, Australia has moved backwards on many crucial sustainability metrics, with greenhouse gas emissions rising [2], biodiversity counts falling [3], plastic pollution spreading [4], wealth inequality widening [5] and now, of course, vast tracts of land burning. The gap between sustainability aspirations and reality is now so stark that it prompts the following question: why have the numerous agreements committing Australia to sustainability these past 30 years not generated far more significant on-the-ground impacts?

A frequent answer to this question is definitional, with a large and diverse normative and empirical literature concluding that sustainability's failure to generate impacts is due to its elasticity and 'slipperiness' [6]. However, from a 'contested concepts' perspective, such conceptual features

are perfectly understandable [7]. When the definitional stakes are especially high, agreement on a concept's analytic core often masks serious disputes over its implementation. From this perspective, 'sustainability' is similar to contested concepts like 'democracy', 'liberty' and 'power'. Embracing sustainability's contested conceptual elasticity, we investigate an alternative explanation for its failure to impact: the need to craft high-level, pluralistic economic, social and environmental compromises that reconcile the deep interdependencies between human and natural systems. We postulate that organisational strategies based on establishing discrete environmental agencies and sustainability committees foreclose precisely the kind of integrated, pluralistic, systems thinking required, misleading expert-based boards into believing that the policies and strategies developed are 'sustainable' despite the limited impact they have on organisational mandates and day-to-day, on-the-ground activity [8]. In summary, we assess the proposition that the 'big disconnect' between the many claims that organisations make to be embracing sustainability and the clear evidence of deteriorating conditions [9] is due to competing conceptions about whether, and if so, how, this needs to be done at the level of the organisational board.

We justify focusing on the board level of analysis by noting how influential boards can be in setting organisational strategies and overseeing and monitoring their implementation. Unlike individuals, board decisions have the potential to significantly alter operational and discursive contexts. When the Australian airline Qantas announced plans to cut onshore waste to landfill by 75% by 2021 and trialled a 'zero waste flight' from Sydney to Adelaide to see what was involved [10], the operational impact was several orders of magnitude greater than any individual's, while the discursive effects obliged other airline industry members to reassess their own waste management strategies. The Qantas decision raises several important questions about board operations generally. Why did the company make the decision at this specific time? Why did it target airline waste and not 'non-essential air travel', given that the latter is a major sustainability issue and contributor to personal, corporate and national carbon footprints? Since we know that the way men and women understand the world differs and influences priority identification and decision-making [11], did the fact that the Qantas board is more gender-balanced than Virgin Australia's board (six men and four women versus nine men and two women) contribute in any way to its decision to lead on airline waste? Since there is also evidence that other demographic characteristics have similar effects [12], we can ask if other types of diversity—age, sexual orientation, ethnicity, disciplinary background, personal values, political leanings—played a role in Qantas's decision?

Overwhelmingly today, boards reflect a homogenous group of white, professional, middle-aged and middle-class individuals, with a significant gender gap in private sector boards, which are composed mostly of men [13]. The lack of board diversity should already be a major issue for public, private and non-governmental organisations, since studies have found that the more diverse governing boards are, the more organisations will engage in corporate social responsibility and avoid fraud, shun aggressive tax planning, undertake transparent reporting and practice corporate philanthropy [11]. There is now considerable evidence to demonstrate that peoples' interpretations of concepts like sustainability are filtered through deep-rooted demographic, class, ethnic, disciplinary, ideological and value positions [14,15], ultimately influencing everyday decisions [16].

As suggested by Rao and Tilt [11], studies are needed that extend the analysis beyond economic and legitimacy metrics to an examination of the relationship between board diversity and sustainability imperatives. In this paper, we answer this call and extend this work by examining, systematically, the existing literature on how board diversity influences the sustainability performance of organisational boards. Currently, to the best of our knowledge, no recent reviews systematically investigate the way that board diversity might influence sustainability performance. Moreover, the reviews that do exist overwhelmingly focus on the private sector, whereas ours extends to the public sector. Finally, in addition to compiling a definitive list of studies, the present review undertakes an analysis of their strengths and weaknesses by comparing and contrasting design and sustainability performance measures. Three basic questions guided our review and synthesis of the literature:

1.  How does board diversity affect the sustainability performance of organisations?
2.  What do we learn from these studies about:

    a.  The characteristics of diversity that have been investigated?
    b.  The way sustainability performance is measured?

3.  What methodologies have been employed to undertake these studies and what are their strengths and weaknesses taken as a group?

## 2. Materials and Methods

For this research, a systematic review with a descriptive and narrative approach is applied to primary studies [17,18]. This method was selected because a descriptive and narrative approach allows a more comprehensive synthesis of different designs, without privileging quantitative or qualitative investigations; important in our case as sustainability performance is measured and reported differently across sectors [19]. In addition, this approach allows us to capture the current state of knowledge (first research question), assess the effects of board diversity on sustainability performance (second research question), and critically review the methodological strengths and shortcomings of studies (third research question). This provides a robust base on which critical insights and the implications of the review can be drawn.

### 2.1. Search Procedure

To search for relevant studies, an electronic and manual search was conducted. The most widely used electronic databases in corporate governance were screened: Scopus, JSTOR, Informit and Web of Science. The descriptors used were: *sustainability* AND *board diversity* OR *board composition*. The combination of those keywords was used to search for both titles and abstracts. The reference lists of the previous review article (i.e., [11]) were also searched manually for the same keywords. A summary of the systematic review process with studies initially selected, main reasons for exclusion, and final pool of studies included in the analysis, is depicted in the flow diagram (Figure 1).

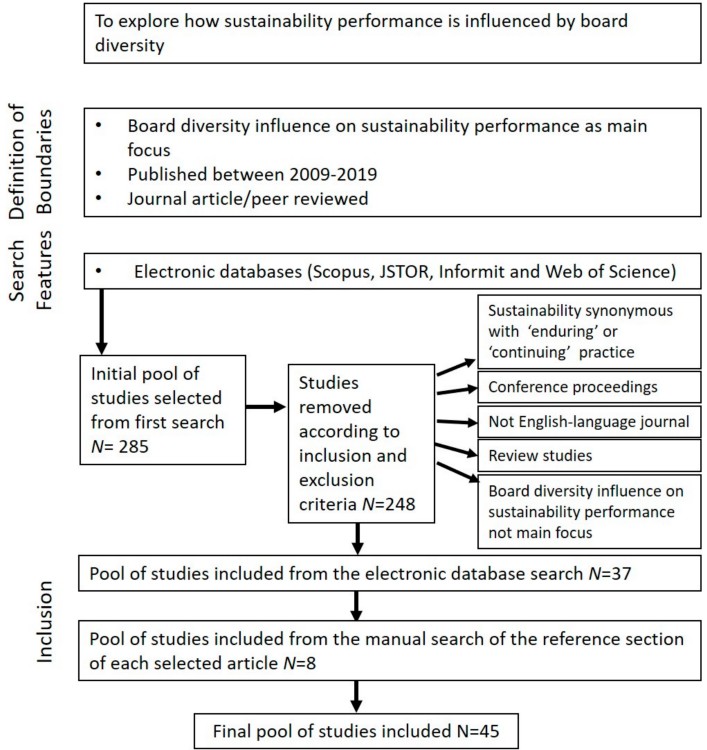

**Figure 1.** Systematic review method (adapted from Civitillo, Juang and Schachner [20]).

*2.2. Inclusion and Exclusion Criteria*

Abstracts and titles were screened to retrieve journal articles relevant to this review. Book chapters and dissertations were excluded because of variable and hard to assess differences in the rigour of the peer review process.

Therefore, to be included in the review, articles had to report on how sustainability was influenced by board diversity, be published in the past decade (2009–2019) (to ensure a more contemporary review of conceptions of diversity and sustainability performance), have the board of organizations as their focal point, and employ a broad interpretation of sustainability performance as integrating economic, social and environmental imperatives. Studies that treated 'sustainability' merely as a synonym for 'continuing' or 'enduring' were excluded, for example, most often in papers relating to financial sustainability. We also only included studies published in international, peer-reviewed, English-language journals that reported some type of data such as panel data, interviews, surveys and observations, regardless of geographical area. Review studies were excluded.

*2.3. Coding Procedure and Data Analysis*

The first author conducted the initial database search of Scopus, JSTOR, Informit and the Web Science and reviewed the titles and abstracts of the studies for potential inclusion. From the list generated, both authors independently reviewed the abstracts of the papers to determine whether they should be included or excluded in analysis. Strong inter-coder consensus on the articles determined relevant to include was found. A decision on disputed studies was taken based on a complete review of the full manuscript.

The first author coded the studies selected using an iterative coding procedure. For the first research question, the following codes were developed:

- Type of diversity investigated (e.g., gender, ethnicity, age, independence, etc);
- Size of the board;
- Sustainability proxy reported and context (e.g., private company, government board); and
- Sustainability performance measure (e.g., annual reports, emissions).

The second question was evaluated by using the main results of each study (e.g., results reported, participant quotes) to determine the effect of board diversity on sustainability performance. Finally, the methodological strengths and weaknesses of each study were assessed, looking at:

- Relationships between authors and participant/companies;
- Presence of a control or comparison group;
- Data gathering and data analysis procedure (e.g., reliability, effect size).

The methodological strength and weakness indicators were informed by previous work on sustainability performance measures [9,19] and on interpretations of diversity [11,21].

## 3. Results

*3.1. Search Results*

The electronic search yielded 285 results. From this electronic search, we retrieved a total of 37 peer-reviewed journal articles after applying inclusion and exclusion criteria. Reference sections of the 37 full-text articles were manually searched for additional articles for inclusion, where a further eight articles were found. A total of 45 articles were subjected to data extraction and thematic analysis, shown in Table 1.

**Table 1.** Results from systematic review with summary information of included articles.

| Reference | Methodological Approach | Diversity Characteristic Investigated | Sustainability Performance Measure |
|---|---|---|---|
| Alazanni et al. 2017 | Quantitative | Gender, board size | Third-party composite |
| Al-Shaer and Zaman 2016 | Quantitative | Gender | Author-generated composite |
| Amran, Periasamy and Zulkafli 2014 | Quantitative | Gender, independence, board size | Author-generated composite |
| Arayssi, Dah and Jizi 2016 | Quantitative | Gender | Author-generated composite |
| Ben-Amar, Chang and McIlkenny 2017 | Quantitative | Gender | Third-party composite |
| Bergman et al. 2015 | Quantitative | Cognitive diversity | – |
| Birindelli, Dell'Atti, Iannuzzi and Savioli 2018 | Quantitative | Gender, independence | Third-party composite |
| Biswas, Mansi and Pandey 2018 | Quantitative | Gender, independence | Third-party composite |
| Boulouta 2013 | Quantitative | Board size, gender | Third-party composite |
| Chams and Garcia-Blandon 2019 | Quantitative | Board size, educational background, gender, independence, age, | Third-party composite |
| Cuadrado-Ballesteros, Martínez-Ferrero and García-Sánchez 2017 | Mixed quantitative and qualitative | Board size, independence, gender | Author-generated composite |
| Cucari, De Falco and Orlando 2018 | Quantitative | Gender, age, independence | Third-party composite |
| Darcus et al. 2016 | Quantitative | Independence, board size, gender | Author-generated composite |
| de Villiers, Naiker and van Staden 2011 | Quantitative | Independence, size | Third-party composite |
| Fakayo and Nakeng 2019 | Quantitative | Gender, independence | Single index |
| Fernandez-Feijoo, Romero and Ruiz-Blanco 2014 | Quantitative | Gender | Author-generated composite |
| Ferrero-Ferrero, Fernandez-Izquierdo and Munoz-Torres 2015 | Quantitative | Generational (veterans, baby boomers, generation X) | Author-generated composite |
| Fuente, Carcia-Sanchez and Lozano 2017 | Quantitative | Gender, independence | Author-generated composite |
| Furlotti et al. 2017 | Quantitative | Gender | – |
| Galbreath 2011 | Quantitative | Gender | Author-generated composite |
| Galbreath 2018 | Quantitative | Gender | Author-generated composite |
| García-Sáncheza, Suárez-Fernándezb and Martínez-Ferrero 2019 | Quantitative | Gender | Author-generated composite |
| Glass, Cook and Ingersoll 2016 | Quantitative | Gender | Third-party composite |
| Hafsi and Turgut 2013 | Quantitative | Gender, ethnicity, experience, age, duality, board size | Third-party composite |
| Haque 2017 | Quantitative | Gender, independence | Single index |
| Issa and Fang 2019 | Quantitative | Gender | Author-generated composite |
| Jizi 2017 | Quantitative | Gender, independence, board, duality | Third-party composite |
| Kassinis, Panayiotou, Dimou and Katsifaraki 2016 | Quantitative | Gender | Author-generated composite |
| Kaymak and Bektas 2017 | Quantitative | Independence, board size, duality | Author-generated composite |
| Kılıç and Kuzey 2019 | Quantitative | Independence, gender, foreign directors, | Single index |

**Table 1.** *Cont.*

| Reference | Methodological Approach | Diversity Characteristic Investigated | Sustainability Performance Measure |
|---|---|---|---|
| Li et al. 2017 | Quantitative | Gender | Author-generated composite |
| Liao, Luo and Tang 2015 | Quantitative | Gender, independence, duality, board size | Single index |
| Mahmood, Kouser, Ali, Ahmad and Salman 2018 | Mixed methods | Gender, independence | Author-generated composite |
| Michelon and Parbonetti 2012 | Quantitative | Independence, community influential representation, duality, | Third-party composite |
| Mohd-Said, Shen and Nahar 2018 | Quantitative | Size, independence, gender | Author-generated composite |
| Nadeem, Zaman and Saleem 2017 | Quantitative | Gender | Third-party composite |
| Ong and Djajadikerta 2018 | Quantitative | Independence, gender, multiple directorships (interlocks) | Author-generated composite |
| Oosthuizen and Lahner 2016 | Quantitative | Gender, ethnicity, independence, non-traditional backgrounds | Third-party composite |
| Ortiz de Mandojana and Alberto Aragon-Correa 2015 | Quantitative | Director interlocks | Single index |
| Post, Rahman and McQuillen 2015 | Quantitative | Gender, independence | Third-party composite |
| Post, Rahman and Rubow 2011 | Quantitative | Gender, age, educational background | Third-party composite |
| Safdar Sial, Zheng, Cherian, Gulzar, Thu, Khan and Vian Khuong 2018 | Quantitative | Gender | Author-generated composite |
| Seto-Pamies 2015 | Quantitative | Gender | Author-generated composite |
| Shoham, Almor, Mook Lee and Ahammad 2017 | Mixed methods | Gender | Third-party composite |
| Tamimi and Sebastianelli 2017 | Quantitative | Board size, duality, gender | Third-party composite |

### 3.2. The Effects of Board Diversity on the Sustainability Performance of Organisations

Most studies analysed in this review find positive correlations between sustainability performance and board diversity. Predominantly, studies consider how the representation of females on a board influences sustainability and, in these studies, debate remains regarding the extent to which the inclusion of females on a board can influence outcomes. Often, researchers advocate for a critical mass of representation of women on a board. Though, Birindelli et al. [22] counters this argument and calls for a greater emphasis on gender-balanced boards, citing Schwartz-Ziv's work that balanced boards demonstrate greater communication and effective problem-solving capacities.

The continued low numbers of females on boards generally was often highlighted (e.g., Issa and Fang, [23]), which Kılıç and Kuzey [24] note may be the reason why they find no relationship between women on boards and carbon emission disclosures by companies in Turkey. Their findings support Seto-Pamies [25], who found a lack of representation of women directors (7%) on boards in their sample population of Global 100 companies; as well as Fernandez-Feijoo et al. [26], who suggest that cultural context may explain why this phenomenon exists.

The dis/enabling aspects of the geographical, cultural and policy context on representations of diversity on boards is highlighted in several studies. Mahmood and Orazalin [27] found that cultural context influenced women on boards in their study in Pakistan where traditional ideas of gender roles were prevalent (similar to [23]). Fakoyo and Nakeng [28] highlight policy as an important enabling factor for the integration of diversity onto boards in their South African study, while Fernandez-Feijoo et al. [26] note the importance of policy when they show a significant negative relationship between the proportion of women on company boards and a country's relative gender equality. Yet the positive impact of women on boards continues to be highlighted no matter the cultural context (Shoham et al. [29]). Other scholars have investigated the difference that diversity on boards can have according to industry. Li et al. [30] found that the environmental policies of industries with greater *Pollution Creation Likelihood* are more positively influenced by women on boards. Similarly, Post et al. [31] focused on the U.S. oil and gas industry to "reveal that as the relative representation of women on the board increases and as the number of independent directors grows, firms are more likely to form renewable energy alliances".

While most studies focused on the presence or absence of females on boards, some considered how gender diversity on boards may influence a company's sustainability performance. Al-Shaer and Zaman [32] find evidence that sustainability reporting quality is higher when boards contain *independent* women on boards. Others considered how the characteristics of diversity are not homogeneous. In their US-based study, Cuadrado-Ballesteros et al. [33] examine how board characteristics may work together to influence the sustainability performance of companies using complexity theory. They advocate that "a female director has more characteristics than her gender" (p. 539). Similarly, Furlotti et al. [34] consider possible self-schemas of individuals as an influencing behaviour, while Galbreath's [35] study on attention-directing structures finds that "women may have greater impact on proximal team effects (e.g. group dynamics such as debate and interaction norms) than on distal effects (e.g. firm performance)" in environmental scanning (p. 753). Darus et al. [36] considers *why* women influence decision-making, although the quantitative methodologies employed limited their capacity to offer rich interpretations of these relationships.

While gender and independence were the most commonly investigated categories of diversity, a few studies investigate ethnicity, educational background and age (e.g., [32,37–42]). Ferrero-Ferrero et al. [43] examine generational diversity (defined by the year that someone is born) on boards of directors and CSR, and while no evidence of a direct effect of generational diversity on CSR performance (measured using Asset4 data) was found, they do find evidence for their hypothesis that *generational diversity positively affects CSR performance by means of CSR management quality*. While not focused on generational difference, Chams and Garcia-Blandon [44] also consider the impact of directors' age on sustainability performance to find a curvilinear relationship between age with sustainability

performance, whereby sustainable practices first increase with age but then decrease as the age of directors rises.

Investigations into other elements of diversity are underdeveloped in this literature, which may be in part due to the general homogeneity of boards. Some researchers who consider other factors include Chams and Garcia-Blandon [44], who find no relationship between the educational qualification, independence or duality of board of directors with sustainability performance. In comparison, Oossthuizen and Lahner [45], in a study of South African companies, find positive (albeit not statistically significant) relationships between board members' ethnicity, non-traditional background, gender and independence, similar to Ortiz de Mandojana and Alberto Aragon-Correa's [46] findings. In Bergman et al. [47], the influence of cognitive diversity to sustainable decision-making is investigated. They find that the cognitive frames of board members tend to privilege economic over environmental and social issues and conclude that currently, sustainability management issues play a minor role in top decision-makers' cognitive frames and strategic landscapes. However, the researchers do not disclose the composition of boards studied nor the identity markers of their directors, and therefore the way cognitive frames may be shaped by individual identities goes unexplored.

While not strictly a characteristic of a board member, several studies investigate the size of the board and a correlation to sustainability is generally found. For instance, Chams and Garcia- Blandon [44] find a statistically significant relationship, Kaymak and Bektas [48] find a positive relationship and Arayssi et al. [49] find that the larger the board, the greater the sustainable performance of companies. Overall, the results of the studies when considered collectively demonstrate that a board of directors with a diverse make-up of people can have a significant impact on the sustainability performance of a company.

*3.3. Characterising Diversity*

Overwhelmingly, diversity in our reviewed studies is conceived through a gendered lens, where the proportions or percentages of women (gender) on boards and their influence on corporate governance is considered. However, independence, board size, duality and interlocks are also commonly investigated. A minority of studies explore generational difference, age, educational background and cognitive diversity as markers of diversity on boards of directors and how these influences corporate governance.

3.3.1. Gender

In all studies, gender diversity is defined in the binary of male/female. Standard means of calculating gender include the percentage or proportion of females on a board, or employing Blau's diversity index [23,24,32,37,40,42,43,49–51]. Shoham et al. [29] offer one of few studies to consider the implications of using binary language, in particular masculine language, in the recruitment of women to company boards. They explore how grammatical gender markings work to reinforce binaries and how this might influence the appointment of women onto a board and the relationship to environmental sustainability.

In some studies [22,23,37,50,52], critical mass is considered in interpretations of diversity and what constitutes 'representation' on a board [22,32,33]. Overall, there are no reports of any boards with more women directors than men.

While numbers of women remain comparatively low across countries, researchers have investigated how women influence governance on boards of directors in organisations all over the world [23,25,28,35,50,51,53]. In all of these studies, the cultural differences that overlay any performance of gender and how women may be able to participate in boards of directors remain largely absent from analyses, with the exception of Alazanni et al. [50].

Most research investigating the influence of women on boards does so in ways that suggest that all women possess the same qualities. Capturing the heterogeneity encompassed within categories of diversity, in this instance what it means to be a woman, is a limited perspective found in the studies included in the review. However, Mahmood et al. [27] discuss the limitations of research that

uncritically accepts social-role theory and the characteristics commonly associated with women. Using interview data, they demonstrate that not all women directors perform in the same way. Furlotti et al. [37] go some way, in acknowledging and investigating this in their study to consider the self-schema of women who are on boards and why they might influence boards in different ways to men. However, they do this through content analysis of CSR-related reports of Italian companies, which arguably is not an adequate methodological approach to investigate a concept related to an individual's experience. Similarly, Glass, Cook and Ingersoll [54] take a partially intersectional approach by considering the proportion of women on the board, the number of interlinks women board members hold, and the interactive and cumulative effects of women CEOs and gender diverse boards.

Overall, the depth of analysis of the studies in how characteristics of diversity are represented, measured and correlated remains largely quantitative. The generalized findings, while demonstrating macro trends, fail to highlight the qualities or attributes of women, or the discrete ways in which gender dynamics play out in board rooms to influence sustainability performance. While connections are made to the relevant literature, purporting why a change may have occurred, the data included in studies (to which [27] is an exception), give little indication into *why* women on boards have an effect or not.

### 3.3.2. Independence, Board Size, Duality: Markers of Diversity

While studies overwhelmingly used diversity synonymously with gender, there were a number that consider the independence of directors, director interlocks, the duality of CEOs and the size of the board [24,31,36,55]. Chams and Garcia-Blandon [44] investigate a range of attributes in their study of diversity on boards and the influence on sustainability performance, including the highest qualification and discipline of board members along with other more commonly reported variables including gender, board size and independence. Fuente et al. [39] refer to race and ethnicity as markers of diversity in their study, although their actual research focuses on the influence of women and independence representation on boards.

Oossthuizen and Lahner [45] pose the question: 'who should the board members be in terms of composition and characteristics?' They seek answers by considering gender, ethnicity, independence and director background. This information comes from annual reports and *Who's Who SA and Business Week.* The authors find that boards of companies listed on the Socially Responsible Index (SRI) are more diverse than a control group of companies. Ethnic minority directors make up 37% of boards for the SRI sample, while they comprise 29% of non-SRI listed companies. Michelon and Parbonetti [56] consider the independence of directors, CEO duality and the representation of influential community members on the board in their study of sustainability disclosures, obtained through annual reports. The inclusion of influential community members is a unique area of investigation, not replicated in any other study in this review.

Other unique approaches to characterizing diversity are the Ferrero-Ferrero et al. [43] and Bergman et al. (2015) studies. Ferrero-Ferrero et al. [43] use BoardEx data to determine the 'generational diversity' of boards, while Bergman et al. [47] consider the cognitive diversity that exists in boards of directors in Finland, explaining this as "the capacity of human cognition relative to the requirements of information environments in which the individuals perform" (p. 163). The scholars found varied effects on sustainability performance.

Overall, the characterization of what constitutes diversity remains largely situated within the category of gender and is in support of Ghauri and Mansi's [21] claim that in the "diversity seems to have been overshadowed by the narrow definition of gender representation and specifically female representation in organisational management and above levels." These researchers call for conceptions of diversity to shift from narrow interpretations towards understandings of *deep diversity,* which they claim includes ethnicity, age, disability and sexual orientation.

In addition, the ways in which categories of diversity were understood in the studies reviewed in this paper were often problematic. Categories of diversity were conceived largely as homogenous

groups, which were assumed to guarantee certain qualities in those inside them. Many of the papers [51,57] drew on understandings from social role theory that allocates discrete differences in social behaviour and personality traits to, for example, males and females. While there may be broad generalizable differences, a greater appreciation of directors as individuals with a complex identity of which gender is but one part is required to deepen our understanding of how certain *qualities* of individuals interact together in decision-making forums.

*3.4. Measures of Sustainability Performance*

As discussed previously, an inclusion criterion for our studies was that sustainability be understood in terms of economic, social and environmental performance. Within the papers reviewed, researchers generally approached sustainability from one of two perspectives: either as a business imperative to be responded to, or as encompassing larger issues facing humanity that business has a key role in resolving (e.g., [45]). Darus et al. [36] emphasise that organisations have a responsibility to 'take care of the communities where they operate, their employees, their customers, the natural environment in executing their economic activities, and ensure the safety of the products that they offer.' Similarly, Galbreath [52] references the Brundtland definition of sustainable development and considers the performance of each pillar of sustainability—economic, environmental and social—independently. Galbreath [52] acknowledges the difficulty of finding reliable proxies for the environmental and social aspects of sustainability and goes on to explain how and why content analysis of annual reports is appropriate: a common method in the field.

Considerable variability in what constitutes sustainability performance is an important finding of this review. A majority of studies use company disclosure as a proxy for determining sustainability performance. Researchers like Ong and Djajadikerta [58] distinguish between 'hard' and 'soft' disclosure using an instrument developed by Ong et al. [59], recognising that disclosure exists along a continuum which they measure using a scale. Others take a binary approach and score evidence of sustainability performance using disclosure/non-disclosure criterion. Al-Shaer and Zaman [32] use *quality* of sustainability reporting as a proxy measure. Arguably, the presence or absence of reporting is not indicative of reporting quality, as much reporting is 'impression management' [60]; nor can auditing evidence of 'quality', given that most audits simply verify a very selected range of information. Fuente [39] makes a similar claim in focusing on the disclosure of information related to sustainability, suggesting that this is, in itself, indicative of sustainability performance.

While the perspectives that informed researchers' conceptions of sustainability vary, overall the ways of measuring sustainability performance are largely similar. In the reviewed works, three approaches are dominant in measuring sustainability performance: (i) single index proxies based on raw data (i.e., energy use); (ii) author-generated composite proxies; and (iii) third-party composite proxies. The majority of studies with author-generated or third-party composite proxies relied on annual reports or sustainability reports of companies for information.

3.4.1. Single Index Proxies

Five studies in our review use raw data of performance to assess sustainability performance. Fakoyo and Nakeng [28] consider how energy usage is influenced by the composition of company boards; Kılıç and Kuzey [24] consider carbon emission disclosures; Ortiz de Mandojana and Alberto Aragon-Correa [46] relate sustainability performance to global warming potential, which they quantify in a sample of electricity generating/transmitting companies; and Haque [61], similar to Liao, Luo and Tang [62], use greenhouse gas emissions (GHG) to measure outcome-oriented carbon performance.

3.4.2. Author-Generated Composite Proxies

A number of studies use annual/sustainability reports or third-party datasets of sustainability related data and then apply various indexes, equations or frames to determine comparable sustainability performance. Ong and Djajadikerta [61] measure sustainability according to company disclosures

and consider total disclosures as well as economic, environmental and social disclosures, basing their assessment of performance on Ong's [60] framework of sustainability disclosure. Li et al. [30] use information on firms' environmental policies available through the KLD database, combined with six items measuring environmental policies related to recycling, pollution prevention and using clean energy. Issa and Fang [23] employ the GRI Guidelines to undertake a content analysis of companies' annual reports, information on websites and sustainability reports. The researchers in this study use a pre-determined format to analyse data for sustainability performance but the analysis of disclosure claims is undertaken independently. Many of these studies discussed the limitations of using publicly available information or information from annual/sustainability reports and noted the risks inherent in self-disclosure [27,53,54,63].

Kaymak and Bektas [49] consider the relationship between corporate governance and CSR and use Transparency International data as a means of measuring CSR. While this does not measure social or environmental indicators directly, the researchers argue that "transparency and disclosure can be considered as a measure of CSR, as the latter is a fluid concept embracing activities that satisfy different interest groups" (p. 557). Arayssi et al. [49] and Fernandez-Feijoo et al. [26] use disclosures as a means of assessing sustainability performance and understand this within a CSR frame. Fernandez-Feijoo et al. [26] use the KPMG International Survey of Corporate Social Responsibility Reporting data compiled on Global Fortune 250 and the 100 largest companies by revenue across 22 countries, which is used as the basis of their analysis about the effect of women on boards on sustainability reporting.

### 3.4.3. Third-Party Composite Proxies

A number of large-scale, well-developed, third-party databases exist that purport to measure sustainability performance. Those commonly employed in the studies reviewed here include KLD, Thomson and Reuters, Dow Jones, Bloomberg and the GRI in various formats [31,44,50,54,56]. Other researchers use Bloomberg's ESG scores in their analyses, including Tamimi and Sebastianelli [64], to determine the sustainability performance of the SandP 500 companies they investigate. The results show that Governance disclosure scores are significantly higher than Social disclosure scores, and Social disclosure scores are significantly higher than Environmental disclosure scores. Nadeem et al. [40] use Bloomberg's ESG score to measure sustainability disclosure and include the environmental categories: water consumption, energy use, wastes management and greenhouse gas emissions. However, the availability of relevant corporate environmental and social information was frequently noted as a major challenge for researchers [22,29,37,57].

Overall, the variety of approaches to determining sustainability performance means that any meta-analysis is likely impossible. Rather, the variety of approaches employed highlights the theory-dependent nature of inquiry. While approaches using single indexes as a proxy ensure that company comparisons are based on like-for-like data, the narrowly defined fields employed, such as energy usage or carbon emissions, do not capture the entire range of a company's sustainability performance, and thus any pernicious intra-environmental or environment-social trade-offs. The variety of author-defined proxies highlight the point made in 2002 by Burritt and, notwithstanding the efforts to standardise corporate sustainability reporting in the GRI, reiterated again by Ong and Djajadikerta [58], that "despite the various methods used in prior research studies, the lack of a standardised reporting framework has hindered comparison of sustainability information". Analyses involving third-party composite data experience the same challenges as presented in the author-defined proxies. The standardization of data sources, the sustainability performance measures employed, and the statistical packages used to ultimately determine sustainability performance are rarely comparable across studies. Of most concern is the reliance on company disclosure to measure sustainability performance. While there were three common approaches identified in the studies reviewed, underpinning each of these approaches was trust in companies to disclose relevant information, either to a third party or through annual/sustainability reporting. The now large literature on corporate reporting as 'impression management' (e.g., [60,65]) highlights how problematic such assumptions are.

### 3.5. Methodological Strengths and Weaknesses

From the studies reviewed, many utilize publicly available organisational information to run quantitative assessments about sustainability performance. While a practical and logistically feasible methodological approach, there are several reliability questions that should be considered. How are each of the companies in the datasets collecting data and are these methods comparable? For example, Fakoyo and Nakeng [28] use publicly available information of 28 companies to run multiple regression analyses to determine relationships between women on boards and energy use. While energy use is a seemingly straightforward indicator, how are the companies measuring energy use? How can the researchers be sure that each approach is comparable to permit between-company comparisons?

Many studies in this review use proxies (i.e., disclosure) to determine sustainability performance, some of which indicate that boards of directors are aware of potential issues related to the use of proxies (such as the presence/absence of sustainability reporting, transparency and company self-assessments) as measures of sustainability. For example, in Mahmood and Orazalin's [27] study, participants highlight that "commitment toward sustainability and greater disclosure of sustainability are two different things" (p. 207). This is a consideration across all studies, where the measures of sustainability rarely seem to reflect company-supplied data of sustainability indicators across environmental, social and economic domains. For example, many studies relied on observations of data from listings. The accuracy and reliability of the data that most studies are drawing on come with no guarantees. The Thomson Reuters ASSET4 website explicitly states, for example, that:

Information within the profile may have been supplied by a variety of different sources and while SRI-CONNECT makes an effort to ensure that any information that we ourselves submit to profiles is accurate and sourced from an appropriate place, neither we nor the organisation that the profile is about can give any warranty as to the accuracy or completeness of information submitted by others.

However, scholars such as Biswas et al. [57] argue that the data is of appropriate quality and is credible. Glass et al. [54] make the point that independent information supplied by KLD (for example) is more credible and reduces the social desirability bias that can be found when annual and sustainability reports are used as a source of data.

There was a tension in the studies reviewed in terms of the credibility and trustworthiness of companies' self-generated reports. For example, Alazanni et al. [50] note as a limitation of their study that data is based on the annual reports of the companies in their sample group and no third-party quality assurance to the data was available. Darus et al. [36], too, acknowledge this limitation: "The selection of this medium [annual and sustainability reports] of CSR reporting was predicated on the notion that the reports possess a degree of credibility and that the contents are not subject to the risk of other interpretations and distortions" (p. 273). Finally, Fuente et al. [39] highlight that "[they] believe that not considering GRI guidelines or focusing only on quantification of the number of GRI indicators included in the CSR report is an important limitation associated with the fact that companies only incorporate indicators that highlight their best CSR performance" (p. 743).

The methodological approaches of reviewed studies are overwhelmingly quantitative and field-based. A strength of the quantitative approach is the ability to control for other variables such as company size, return on equity and, in some cases, bank leverage (such as [22]). However, within the quantitative studies, the use of control groups is limited and the use of simulation methods to allow for greater control and comparison of variables is absent. Michelon and Parbonetti [56] are an exception, including a control group in their GRI-informed content analysis of the influence of board diversity. Oossthuizen and Lahner [45] also utilise sampling to incorporate a control group and draw on companies from the FTSE/JSE All Share Index with a Social Responsible Investment Index and a second population that are not listed in the SRI.

Researchers took various approaches to collecting data, that included different databases, different geographical regions as well as different industries; however, the analytic processes are largely similar. As noted by Cuadrado-Ballesteros et al. [33], "the most common methodology used by researchers is the multiple regression analysis, a symmetric test that reports the net effects of some variables

on a dependent variable, considering a set of other independent variables" (p. 529). While the quantitative studies are able to provide macro-level trends, limitations include the inability to provide a rich interpretation of why these trends are apparent. For example, Arayssi et al. [49] find that the participation of women on boards of directors positively influences ESG disclosure and speculate that "women directors seem to promote social agenda in the boardrooms . . . " (p. 392). However, there is no data from the research that can support this claim, other than to infer the relationship.

Only three studies in our review draw on qualitative methods or theory. Mahmood et al. [27] demonstrate the value of qualitative methods in the nuance they reveal in how diversity on boards can vary in different situations. Shoham et al. [29] apply a mixed methods approach to collecting data on the influence of board composition on sustainability performance, using interviews with board directors to inform their interpretations of the quantitative results. Cuadrado-Ballesteros et al. [33] employ a qualitative comparative analysis, which is predominantly a quantitative assessment of the relationship between the variables under investigation. However, their approach is innovative in analysing the results through the lens of complexity theory. This approach allows the researchers to problematize the correlations obtained and consider the relationality of board characteristics and how they may work together to produce different outcomes.

Overall, the methodological approaches taken in the studies reviewed allow for generalisable understandings of how some characteristics of diversity influence sustainability performance. While only a few studies interrogate these findings at the micro-level, there are approaches being used to enable a more nuanced understanding of *how* diverse compositions of board members influence sustainability performance. Taken together, and as noted by other researchers, there is a need for further research that provides more descriptive and comparative accounts of the full range of diversity on boards for sustainability (noted also in [45]) supplemented by longitudinal studies [52].

## 4. Discussion

There are several important practical and theoretical findings from this review. We find evidence to support disrupting traditional compositions of boards that tend to represent only white, male, middle class, urban identities as a means of achieving sustainable outcomes in organisations. Yet, the variability in the studies reviewed around definitions and methodology make it difficult to claim what kinds of diversity matter and the level to which outcomes can be improved. Therefore, while there is evidence that changes to board composition can contribute to sustainable outcomes, the way diversity and, to some extent, sustainability performance, is theoretically constituted, currently undermines policy and strategic efforts to enforce such a radical means of ensuing sustainable outcomes.

On this note, we found the existence of a huge range of indicators in use by researchers to measure sustainability performance. While this variability reflects the multiple interpretations of sustainability in use and the complexity of capturing it within competing frameworks, the variability in indicators raises questions regarding construct validity. That is, how accurately are these studies measuring what it is that they claim to be testing? The majority of studies in this review relied on corporate disclosure as a proxy for sustainability performance. While research suggests that disclosure is an appropriate measure of sustainability [66], disclosure relies heavily on the self-reporting of the organisations under investigation, while the methodologies used by individual organisations to gather data for reported indicators are often not comparable across organisations or industry sectors. These observations raise important methodological questions regarding how to operationally define sustainability in the first instance. For example, can 'sustainability' be unambiguously defined to obtain a proxy measurement that allows for comparability? Should it be left up to individual researchers to undertake this task or is there a need for a global commission to do it? If so, who should participate in such a commission and with what influence?

Our findings also reveal the troublesome ways that 'diversity' has and continues to be defined, especially in the management literature where corporate board diversity studies are predominantly published. This research frequently conflates the broad idea of diversity with the narrower inclusion

of women on boards. We support those who argue for interpretations of gender that are more inclusive and include contemporary understandings of gender identity that recognise non-binary identifications [67]. We suggest that more research is needed that investigates other forms of diversity (i.e., ethnicity) and its influence on sustainability performance [68,69]. As Glass [54] notes:

... future research could consider the role of other types of diversity on organizational policy and practice related to the environment. Though much less scholarship has considered the effect of racial/ethnic diversity as compared with gender diversity on organizational practice, scholarship on that topic suggests that racial/ethnic minority leaders bring diverse professional experiences and perspectives to leadership positions. (p. 508)

To this we would add the importance of investigating diversity in political perspectives along the left–right spectrum and, building on the extensive work by Schwartz [70] and his colleagues, on personal values. There is also a need for 'intersectional' research that examines how the complex characteristics of individual board members in terms of gender, ethnicity, age, discipline, cognitive capacity and political and personal values cumulatively influence board decisions, including decisions concerning its sustainability responsibilities, reporting and practices. With regard to the former, it would be hard to underestimate the disruptive impact, for example, of introducing legislation that mandated that the appointment of directors to all boards, public and private, required organisations to balance the number of those with egocentric values (linked to 'achievement') equally with those holding altruistic (linked to 'benevolence') and biospheric values (linked to 'universalism'). With regard to the latter, mandating that boards provide evidence of high intersectionality would also have profoundly disruptive impacts, given how homogeneous most boards currently are.

While the studies show evidence of relationships between various attributes of board members in influencing sustainability, there are few studies that robustly work towards explaining why these associations exist. Most studies use various forms of regression analyses and while this methodology is useful for macro-patterning, it does not provide interpretations on why or how these patterns have occurred. Qualitative and mixed methods approaches are largely absent from the body of research in this review and future research using these approaches could help to address this current gap. There are also opportunities to employ simulations of board decision making which, by controlling for many of the intervening variables, could assist in assessing the specific contribution of designated independent variables on the dependent, sustainability, variable.

In addition, this study did not find any research solely focused on investigating the role of diversity on boards of governmental agencies and public sector organisations, with the exception of Sangle [71], whose focus remained on CSR. The literature is currently dominated by investigations into the board diversity of corporate, business and private sector organisations. This is a large gap in the international literature base in urgent need of addressing given the important role that public sector boards play in environmental, social and economic decision making.

This review has demonstrated the many ways in which board composition may significantly influence sustainable outcomes. To improve organisations' sustainable decision-making, we argue that disrupting the status quo of board composition is one evidenced way to achieve this. While we have identified several important areas of research that need addressing, we argue that this review indicates that diverse board composition is a means of embedding sustainable decision-making into organisations, worthy of further investigation. While many unknowns remain regarding the quality or quantity of impacts that may be realised through changes to board composition, what is known is that maintaining the 'status quo' is no longer viable. Innovative solutions must be found that entrench decision-making for sustainability in organisations which work to create a *new* status quo and contribute to the system-wide disruption so urgently needed across the public and private sector.

## 5. Limitations

As with all studies, ours is not without limitations. First, a general lack of consensus among organisations and scholars as to what constitutes sustainability, its disclosure and its performance

as well as the complex relationships between each of these dimensions, generates an extremely heterogenous set of studies that significantly complicates our understanding of the influence of board diversity. In addition, the lack of qualitative research of organisational sustainability performance and the role of management, in particular boards of directors, has meant that our findings in this paper can only be tentative. Finally, it is acknowledged that the scope of this paper is limited to the search terms used and its relevance, to the time at which it was conducted.

**Author Contributions:** Conceptualization, F.G. and K.B.; methodology, K.B.; validation, F.G. and K.B.; formal analysis, K.B. and F.G.; investigation, K.B.; writing—original draft preparation, K.B.; writing—review and editing, K.B. and F.G.; supervision, F.G.; project administration, K.B. All authors have read and agreed to the published version of the manuscript.

**Funding:** This research received no external funding.

**Conflicts of Interest:** The authors declare no conflict of interest.

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
