# Peer review of "Disrupting the Status-Quo of Organisational Board Composition to Improve Sustainability Outcomes: Reviewing the Evidence"

_sustainability, doi:10.3390/su12041505_

Round 1

Reviewer 1 Report

Thank you for the opportunity to review and comment on this paper. In my opinion, it deals with a very important and up-to-date issue of organisational performance, indicates the most important aspects of organizational sustainable development and the role of board diversity in this field. The article presents the complexity of the studied issues in a very interesting way and identifies research needs. The applied research methodology, allowing for comprehensive analysis of the board diversity role in the sustainable development of organisations, was also clearly presented. Similarly, the results of the analyses successively present findings referring to individual research questions.

However, I have two suggestions to improve this article. The first one  regards to the names of 3.2, 3.3, and 3.4 points. I would suggest to paraphrase them instead of using research questions as their name. In addition, it is needed to indicate clearly in the discussion section: ​​1. theoretical, and 2. practical values of the results. In my opinion, suggested corrections will lead to strengthen the scientific side of the reviewed article. After consideration of those changes, the article could be published in “Sustainability”.

Author Response

Point 1 - The first one  regards to the names of 3.2, 3.3, and 3.4 points. I would suggest to paraphrase them instead of using research questions as their name. 

Thank you for this comment. We have shortened each of these subtitles to now read:

3.2 The effects of board diversity on the sustainability performance of organisations

3.3 Characterising diversity

3.4 Measures of sustainability performance

3.5 Methodological strengths and weaknesses

Point 2 - indicate clearly in the discussion section: ​​1. theoretical, and 2. practical values of the results. 

We have addressed this comment by including the following clarifications to paragraph 1 of the discussion:

There are several important practical and theoretical findings from this review. We find evidence to support disrupting traditional compositions of boards that tend to represent only White, male, middle class, urban identities as a means of achieving sustainable outcomes in organisations. Yet, the variability in the studies reviewed around definitions and methodology make it difficult to claim what kinds of diversity matter and the level to which outcomes can be improved. And so while there is evidence that changes to board composition can contribute to sustainable outcomes, the theoretical constitutions of diversity and to some extent, sustainability performance, currently undermine policy and strategic efforts to enforce such a radical means of ensuing sustainable outcomes.

Reviewer 2 Report

The writing is clear and precise but there is a little repetition between sections that is redundant. This is very interesting work so any additional points would add value rather than repeating already mentioned detail. The work is important and well presented.

Author Response

Thank you for the encouraging comments on our paper. We thank the reviewer for the time and energy invested in conducting this review.